# Association between Physical Activity, Sedentary Behavior, Satisfaction with Sleep Fatigue Recovery and Smartphone Dependency among Korean Adolescents: An Age- and Gender-Matched Study

**DOI:** 10.3390/ijerph192316034

**Published:** 2022-11-30

**Authors:** In-Whi Hwang, Ju-Pil Choe, Jeong-Hui Park, Jung-Min Lee

**Affiliations:** 1Graduate School of Physical Education, Kyung Hee University (Global Campus), 1732 Deokyoungdaero, Giheung-gu, Yongin-si 17014, Gyeonggi-do, Republic of Korea; 2School of Public Health, Texas A&M Health Science Center, 212 Adriance Lab Rd., College Station, TX 77843, USA; 3Sports Science Research Center, Kyung Hee University (Global Campus), 1732 Deokyoungdaero, Giheung-gu, Yongin-si 17014, Gyeonggi-do, Republic of Korea; 4Department of Physical Education, Kyung Hee University (Global Campus), 1732 Deokyoungdaero, Giheung-gu, Yongin-si 17014, Gyeonggi-do, Republic of Korea

**Keywords:** smartphone dependency, physical activity, sedentary behavior, satisfaction with sleep fatigue recovery, a national survey

## Abstract

The purpose of this study was to identify the association between physical activity (PA), sedentary behavior (SB), satisfaction with sleep fatigue recovery (SSFR), and smartphone dependency in South Korean adults. We analyzed data from the National Health and Nutrition Examination Survey 2020 data. We selected participants who answered Internet addiction-related questions as “Very much” (*n* = 241) and answered Internet addiction-related questions as “Not at all” (*n* = 241) in the questionnaire. The participants were matched by age and gender, then divided into two groups. Between the two groups, there were considerable differences in the number of days participating in moderate to vigorous PA (5 days or more, *p* = 0.01), the number of strength training days (1 day, *p* = 0.02), the number of light PA days for more than 60 min (every day for the last 7 days, *p* = 0.01), and the SSFR over the past 7 days (*p* < 0.05). Additionally, the mean smartphone usage time and mean sedentary behavior time between the two groups showed significant differences. The study demonstrated that there were significant associations between PA, SB, SSFR, and smartphone dependency among Korean adolescents matched by age and gender. Additionally, this study highlights the importance of increasing overall PA and number of days participating in MVPA, decreasing SB time and smartphone usage time could reduce the incidence of smartphone overdependence.

## 1. Introduction

The smartphone usage rate continues to increase worldwide every year, and the number of smartphone users has reached more than half of the world’s population [1]. Especially, the usage of smartphones during adolescence is higher than ever before, because technological advancements make it possible to watch broadcasts, take classes, and play games through smartphones that were previously impossible [2]. According to the national survey, in adolescents living in the United States, increased from 35% in 2011 to 81% in 2019 [3], and about 97% of Swiss adolescents [4] and 84% of Japanese adolescents reported that they own a smartphone [5].

Beyond just the function of the telephone, comprehensive service (i.e., freely using the Internet, communicating, and recording) in smartphones is providing better accessibility and convenience day by day [6] and have established omnipresence in our daily lives [7,8,9]. While widespread smartphones have positive aspects, it does not always bring beneficial changes due to the excessive use of smartphones [10]. According to the results of the 2020 Smartphone Overdependence Survey by the Ministry of Science and Information and Communication Technologies (ICT) of Korea, the percentage of smartphone users at risk of overdependence was 23.3%, up 3.3% from the previous year, of which the percentage of adolescents was particularly high at 35.8% [11]. This means that as the penetration of the Internet increases, students are naturally exposed to video terminals such as laptops, tablets, and desktops for a long time [12,13].

Previous studies revealed that the increase in smartphone use caused users to have problems such as overdependence and addiction [14,15], this is a bigger problem for adolescents who are in the growth period [16]. The drawbacks include physical problems such as scoliosis, carpal tunnel syndrome, and dry eye, along with psychological disorders such as depression, anxiety, memory loss, and sleep disorders with poor quality of sleep [17,18,19,20,21,22]. Therefore, the National Information Society Agency classified smartphone addiction symptoms; as (1) smartphone use as the most important activity in daily life (salience), (2) smartphone usage time becomes difficult to self-regulate (self-control failure), (3) a state in which there are serious consequences for conflict with people around, physical discomfort, and difficulties in the home, school, and work–life [23]. Although smartphone overdependence is not an official mental disorder, it is an important one of the pressing clinical problems facing society today [24].

Excessive smartphone use interfered with physical activity (PA) and included impairment of social and emotional functioning [25]. Many studies have reported that increased sedentary time is associated with several health problems such as impaired glucose absorption, increased food intake and waist circumference, and mortality risk, and described that it affects blood pressure problems and exercise habits [26,27,28,29]. Sedentary behavior (SB) induced by smartphone use can be defined as functions such as phone calls, text conversations, and Internet [30]. Specifically, Tammelin and colleagues investigated that high Internet and computer use and measured SB in Finnish adolescents aged 15 to 16 years old was associated with high body mass index and low levels of PA [31], which means that adolescents who frequently use smartphones are associated with various health problems, including obesity, metabolic syndrome, and decreased cardiorespiratory health [32].

In a technologically advanced society, poor sleep quality has emerged as a public health problem [33]. Previous studies have reported that problematic internet use causes problems such as insomnia and poor sleep quality [34,35,36]. Among those studies, Crowley and his colleagues reported that using a smartphone or tablet PC before bedtime could disrupt circadian rhythms due to the light emitted from the screen [36]. Excessive smartphone use delays the onset of sleep and disrupts sleep patterns [37], so problematic Internet use in adolescents causes unsatisfactory sleep quality and poor academic performance [38]. Sleep satisfaction is a very important factor related to the quality of life for adolescents who are growing up to lay the foundation for their overall life [39,40].

Although there are many previous studies related to smartphone overdependence, none of the studies have explained the relationship between PA, SB, and satisfaction with sleep fatigue recovery (SSFR) according to smartphone dependency. In addition, most of the studies have a limitation in that they explained the relationship between smartphone addiction and mental health. Therefore, the primary aim of the current study is to examine the association between problematic smartphone use and PA, SB, and SSFR in adolescents aged 12 to 18 years old.

## 2. Materials and Methods

### 2.1. Study Participants

This study was analyzed using the raw data from the 16th (2020) Youth Behavior Online Survey which is a government-approved statistical survey that has been performed annually since 2005 and conducted by a Korean government agency to determine the status and level of health behaviors of Korean adolescents based on the National Health Promotion Act (Article 19) by the Korea Disease Control and Prevention Agency, Ministry of Health and Welfare of South Korea, and Ministry of Education, Science, and Technology (approval number 117058).

The subjects of the survey were 54,948 youth enrolled in 793 middle and high schools in Korea (398 middle schools, 395 high schools), and the participation rate was 94.9%. After distributing guidelines per participant, the teacher explained the necessity of the survey and how to participate, and all participants involved in the study surveyed completed informed consent online before commencing the survey.

### 2.2. Anthropometrics

The assessment of height and weight was based on self-reported data. As the participants were Korean adolescents, the age range was 12 to 18 years old and height, weight, and BMI were classified by rounding to two decimal places. Additionally, BMI was calculated by dividing the weight (kg) by the square of the height (m^2^).

### 2.3. Measures

#### 2.3.1. Smartphone Dependency Scale

The group was divided by considering Internet addiction in the questionnaire. The answers to 10 questions consisted of: (1) not at all, (2) no, (3) yes, and (4) very much. The questionnaire is as follows; (1) I failed whenever I tried to reduce my smartphone usage time, (2) It is difficult to control smartphone usage time, (3) It is difficult to keep an appropriate smartphone usage time, (4) It is difficult to concentrate on other tasks when the smartphone is next, (5) Smartphone thoughts(ideas) do not leave my head, (6) I feel a strong impulse to use my smartphone, (7) There is a health problem because of smartphone use, (8) Because of the smartphone, I have experienced severe conflicts in my family, (9) Because of the smartphone, I have experienced severe conflicts with friends (colleagues) and social relations, and (10) There is difficulty in doing business (academic or vocational) due to smartphones. The Cronbach’s alpha coefficient for questionnaire was 0.995.

#### 2.3.2. Group

The participants who answered “not at all” to all of the questions were divided into the normal group (10 points, *n* = 9684) and those who checked the “very much” to almost all of the questions were divided into the overdependence group (37–40 points, *n* = 241). Since the number of students in the overdependence group was small, the number of people in the normal group was matched by age and gender based on the overdependence group, we finally made the normal group (Group 1, 10 points, *n* = 241) and the overdependence group (Group 2, 37–40 points, *n* = 241).

#### 2.3.3. Physical Activity (PA)

PA scores were calculated by answering the following questions; (1) How many days did you do more than 60 min of PA (regardless of type) that caused your heart beat rate to increase? (i.e., walking, low-intensity biking), (2) How many days did you do 20 min or more of moderate to vigorous PA (MVPA) that made your body sweat? (i.e., soccer, basketball, hiking, jogging), and (3) How many did you do muscle strength training? (i.e., push-ups, sit-ups, and lifting weights). The standard of all days was one week.

#### 2.3.4. Sedentary Behavior (SB)

The question to ask SB was ‘How much time (i.e., minutes) do you spend sitting down on average per weekday and weekend (purpose for learning or not)’. The purpose of learning includes classes at school and academic institutes, watching TV or using computers for homework or study, and watching educational broadcasts. Non-learning purpose includes watching TV, playing games, internet, chatting, and sitting while moving.

#### 2.3.5. Satisfaction with Sleep Fatigue Recovery (SSFR)

The degree of SSFR was measured through the questionnaire ‘Do you think the amount of time you have slept for the past 7 days is sufficient to recover from fatigue?’, then they were classified as; (1) very enough, (2) enough, (3) just enough, (4) not enough, and (5) not enough at all.

### 2.4. Data Analysis

All data processing and statistical analysis used SPSS 26.0 version (SPSS Inc., Chicago, IL, USA). The demographic information of the participants was summarized in descriptive. This was examined by the descriptive statistics of the participants who were divided into two groups (i.e., normal and overdependence group) and the participants’ personal information such as gender, age, and anthropometrics (i.e., height, weight, and BMI). Additionally, the independent t-test was utilized to investigate the differences between the questionnaire-assessed smartphone usage time and SB time. 2-way ANOVA and Bonferroni Post hoc tests were utilized to confirm the interaction effect between variables in SB time. For SSFR, frequency analysis was conducted to confirm the response proportion between the two groups. Moreover, multinomial regression analysis was performed to find out the associations between PA, SB, SSFR, and smartphone dependency. Results from this analysis are presented as odds ratios (OR) with 95% confidence intervals and statistical significance set by *p* < 0.05.

## 3. Results

Table 1 shows the demographic participants’ characteristics of students using descriptive statistics (*n* = 482). Demographic characteristics such as gender, age, height, weight, and BMI are presented in numbers in Table 1. Since we matched age and gender, there was only one difference in the number of genders between the groups, and they were all the same age. The results indicated that there were no significant differences in height (*p* = 0.93), weight (*p* = 0.35), and BMI (*p* = 0.24) in the two groups. The adolescents aged 17 years represented the largest group with 23.2%, while the adolescents aged 12 were the smallest group with 4.6% in the age ratio of the students who participated in the survey.

Figure 1 illustrates the comparison of average smartphone usage time using an online survey between two groups. The average smartphone usage time was 611 ± 417 min per week in the normal group, but the participants in the overdependence group were 1002 ± 482 min per week. The results demonstrate a significant difference in the smartphone usage time between the normal and overdependence groups (*p* < 0.001; Cohen’s *d* = 0.87; 95% CI = −471.47–−310.27).

In the same way, Figure 2 shows the average SB time on weekdays and weekends for the two groups. In the cases of sitting for weekday learning (or not) purposes, the mean SB time of the normal group was 441 min for weekday learning purposes and 213 min for weekday purposes not for learning, and likewise, the SB time of the overdependence group was 372, 279 min, respectively. Additionally, in the cases of sitting for weekend learning (or not) purposes, the SB time of the normal group was 232 min for weekend learning purposes and 346 min for weekend purposes not for learning, and similarly, the SB time of the overdependence group was 197 min and 414 min, respectively. Among the four variables, there was no significant difference only in the case that sitting for weekend learning purposes. As a result of conducting 2 way ANOVA, there was a significant interaction effect between groups and sitting purposes, F(3, 1920) = 14.87, MSE = 51,579.75, *p* < 0.001, partial η^2^ = 0.02. Additionally, as a result of the Bonferroni post hoc test, there was no significant difference only between weekend learning purpose and weekday no-learning purpose. M_diff_ = −31.34, 95% CI [−70.01, 7.30], *p* = 0.19.

Figure 3 shows the proportion of participants who responded to the SSFR survey. The proportion of participants who answered that ‘The last 7 days of sleep were very enough for fatigue recovery’ was 16.2% in the normal group and 10% in the overdependence group. Likewise, students who answered enough and just enough were 17.4% and 33.2% in the normal group, but 9.5% and 19.9% in the overdependence group, respectively. In contrast, the proportions of students who answered not enough and not enough at all were 21.6% and 11.6% in the normal group, and 26.6% and 34% in the overdependence group, respectively.

Table 2 is the result of a multinomial logistic regression analysis to demonstrate factors that can find out the associations between PA, SB, SSFR, and smartphone dependency. According to Groups 1 and 2 in Table 2, smartphone overdependence was related to the number of light PA days (more than 60 min per day) in every day for the last 7 days (OR = 0.21; CI = 0.07–0.69; *p* = 0.01), the number of MVPA days in 5 days or more (OR = 4.10; CI = 1.39–12.13; *p* = 0.01) and the number of strength training days in 1 day (OR = 2.60; CI = 1.18–5.72; *p* = 0.02) and SSFR degree (Very enough: OR = 3.43; CI = 1.63–7.24; and *p* = 0.00, Enough: OR = 3.43; CI = 1.63–7.23; and *p* = 0.00, Just enough: OR = 4.99; CI = 2.66–9.34; and *p* = 0.00, Not enough: OR = 2.07; CI = 1.11–3.87; and *p* = 0.02) measured by questionnaires were significant risk factors in predicting smartphone overdependence in adolescents.

## 4. Discussion

With the development of science and technology, the rate of smartphone users is higher than ever, and adolescents’ health risk behaviors due to smartphone overdependence are becoming an issue. Among the adolescent health risk behaviors, lack of PA participation, increased SB time, and SSFR is very important for adolescent health risks and the quality of life. For these reasons, smartphone overdependence among Korean adolescents can cause notable clinical and public health problems. Therefore, we examined the association between the smartphone dependency of adolescents and other health risk behaviors. In the present study, we demonstrated that higher smartphone dependence levels had statistically significant associations with PA participation rate, SB time, and sleep characteristics.

In this study, the number of students who participated in moderate to vigorous PA for 5 or more days over the last 7 days was about 4 times more in the normal group than in the overdependence group (*p* < 0.05). Additionally, it was confirmed that the values of the odds ratio between the number of students participating in PA were 2.23 times, 1.45 times, 1.32 times, and 1.7 times higher in the normal group than in the overdependence group for 4 days, 3 days, 2 days and 1 day, respectively. Similarly, the normal group showed a higher participation rate than the overdependence group in the number of days participating in strength training. In particular, the number of students who participated in strength training activities for 1 day in the last 7 days was 2.60 times higher in the normal group than in the overdependence group. However, contrary to our expectations, the number of students who participated in low-intensity PA for every day for the last 7 days was higher in the overdependence group. This founding indicated that students’ misinterpretation of PA intensity, inaccurate recall, and subjective characteristics of the questionnaire might be influenced on the light PA intensity. The results of the present study demonstrated that participation in PA was significantly different between the normal group and the overdependence group. Several studies show that PA participation was positively associated with improved physical and mental health, and sleep efficiency [41,42,43,44]. It has been highlighted that regular participation in PA can positively affect adolescents’ physical and mental health and help improve academic achievement, cognitive ability, and self-esteem [45,46]. Some studies have also presented the association between Social Network Service (SNS) users and smartphone overdependence and they found using SNS is more often used by individuals with relatively low self-esteem, and it interferes with social interaction [47,48]. As a result, people with low social skills spend more time on social media and develop smartphone overdependence [49]. In fact, Sandhya and colleagues investigated that students who spent time in sedentary behavior to use social media were less likely to participate in moderate to vigorous PA, and conversely, students who less frequently used social media were more likely to engage in moderate to vigorous PA [50]. Therefore, it can be inferred that PA participation contributes to reducing adolescents’ dependence on smartphones.

In SB time, the normal group spent more sitting time for learning purposes during the weekdays than the overdependence group, whereas the time spent sitting for non-learning purposes was more in the overdependence group than in the normal group. From these results, we could speculate that students addicted to smartphones had lower academic interest than those in the normal group and confirmed that the purpose of using the Internet was not for learning but other purposes. This supports a previous study by Smaha, who found that smartphone addiction was negatively correlated with academic performance [51]. Additionally, research by Kantomaa and his colleagues about the associations of PA and sedentary behavior with adolescent academic achievement, found that students with high PA participation rates had 1.58 times higher grade point averages (GPA) than those with low participation rates. Regarding Internet use and video games, students who spent less than an hour per day had a 1.36 times higher GPA than those who spent more than two hours per day and related to SB time, students who sat for 1–2 h were shown 1.35 times higher GPA than students who spent more than 2 h per day [52]. In addition, as mentioned earlier, prolonged SB time is highly associated with obesity, metabolic disease, and cardiovascular disease in adolescents [53,54]. Therefore, this study suggests that appropriate control of SB time is necessary and that active PA should be encouraged for the healthy life of adolescents.

Interestingly, there was a very clear difference between the two groups in SSFR. Compared to the overdependence group, the number of students who answered ‘Very Enough’ to the satisfaction question of recovery from fatigue after sleep was 3.43 times higher in the normal group than in the overdependence group. Wood’s study found that excessive Internet use and screen time at the midnight can affect sleep quality by delaying the onset of melatonin secretion and the delayed sleep onset due to bright light [55]. Having enough sleep is a very important factor in life that affects mental health, along with the findings of Leila and colleagues that sleep quality contributes to emotional regulation and cognitive ability, and is involved in memory performance [56]. Therefore, maximizing sleep efficacy among adolescents who are in the process of growth and development is indispensable for improving their quality of life. The average study hour of Korean adolescents is often between 12 to 16 h, which was 7~11 h longer than the average study hour of adolescents in other countries [57]. Korean adolescents try to relieve their academic stress by using smartphones that are easy to carry and full of stimulating content [7,58]. This leads to sleep disturbance and reduced sleep efficiency, further suggesting that over-dependency on smartphones strongly reduces the quality of life and positive health behavior in adolescence.

The present study revealed several positive strengths. First, none of the previous studies have investigated an association between PA, SB, SSFR, and smartphone dependency using age-, and gender-matched adolescent data between the groups. We also utilized PA participation intensity and duration to examine the association between smartphone dependence rate and PA participation type. Sedentary behavior was divided by time and purpose, which made it possible to examine that sedentary time differed according to the purpose of SB between groups. It also has the advantage of using national representative data set with a large, matched sample size. However, this study has a few limitations. First of all, self-report questionnaires were used to measure PA volumes instead of device-based measures such as an accelerometer, leading to an over/underestimation of true PA volumes. Second, the design of this study is a cross-sectional study, a causal relationship cannot be inferred. Lastly, the participants were only Korean adolescents and did not control for the social economic status for both groups. Therefore, we suggest that other studies in the future will need to consider more valid methods, investigate the diversity of ethnicities, and adequate sample size.

## 5. Conclusions

Adolescence is a critical period directly related to establishing lifelong positive or risky health-related behaviors and those behaviors usually persist during the adult years. Therefore, forming positive healthy behavior during this period is very crucial. Due to rapid technology development, adolescents in their growth and development stages may be exposed susceptible to various physical and psychological risks. Although we cannot draw a solid conclusion from this present cross-sectional study, this finding indirectly suggests that smartphone overdependence affected negative adolescents’ lifestyles and behavior. Smartphone overdependence brought about changes in PA participation rate, SB time, and sleep characteristics that could result in deteriorating health outcomes, including adolescents’ physical and mental health. Thus, the government, schools, and families must provide guidelines for appropriate smartphone usage time to form healthy adolescent behavior. The differences in PA participation, SB time, and SSFR observed in our study indicate that it may be necessary to strictly measure and review the smartphone dependence rate of adolescents for their health.

## Figures and Tables

**Figure 1 ijerph-19-16034-f001:**
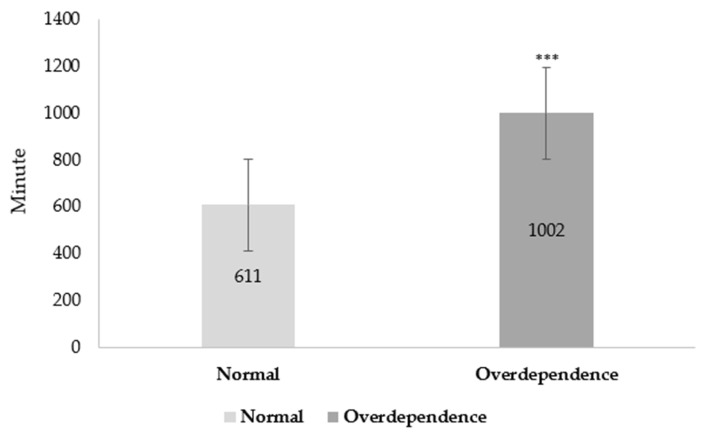
Mean of smartphone usage time (minute per week). The significant differences between the two groups. *** *p* < 0.001, Error bars represent the 95% confidence interval.

**Figure 2 ijerph-19-16034-f002:**
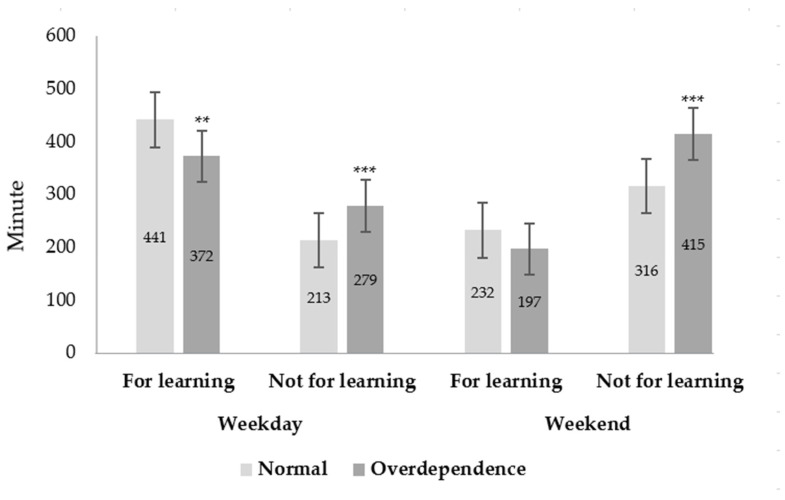
Mean of sedentary behavior time (minute per week). The significant differences between the two groups. *** *p* < 0.001, ** *p* < 0.01, Error bars represent the 95% confidence interval. There was a significant interaction effect between groups and sitting purposes, F(3, 1920) = 14.87, MSE = 51,579.75, *p* < 0.001, partial η^2^ = 0.02.

**Figure 3 ijerph-19-16034-f003:**
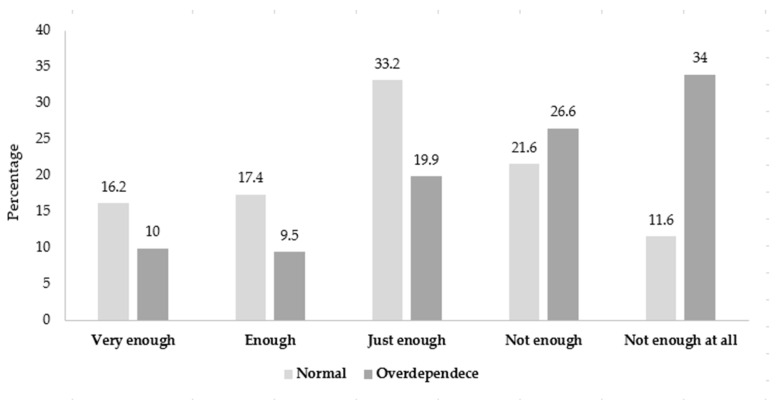
The proportion of participants who responded to the satisfaction with sleep fatigue recovery survey.

**Table 1 ijerph-19-16034-t001:** Descriptive statistics for participants in the general and overdependence groups. (*n* = 482).

Variable			Normal (*n* = 241)		Overdependence (*n* = 241)	
			No. (%)	Mean ± SD	No. (%)	Mean ± SD
Gender	Male		117 (48.5)		116 (48.1)	
	Female		124 (51.5)		125 (51.9)	
Age (year)	12 year		11 (4.6)		11 (4.6)	
	13 year		26 (10.8)		26 (10.8)	
	14 year		36 (14.9)		36 (14.9)	
	15 year		52 (21.6)		52 (21.6)	
	16 year		35 (14.5)		35 (14.5)	
	17 year		56 (23.2)		56 (23.2)	
	18 year		25 (10.4)		25 (10.4)	
Anthropometrics	Male	Age (year)		15.56 ± 1.7		15.58 ± 1.7
		Height (cm)		171.9 ± 8.0		172.1 ± 8.6
		Weight (kg)		64.5 ± 14.2		65.9 ± 14.2
		BMI (kg·m^−2^)		21.8 ± 4.2		22.2 ± 4.2
	Female	Age (year)		15.29 ± 1.7		15.27 ± 1.7
		Height (cm)		161.3 ± 5.3		161.3 ± 5.1
		Weight (kg)		52.9 ± 8.6		53.9 ± 8.3
		BMI (kg·m^−2^)		20.3 ± 2.9		20.7 ± 2.8
	Total	Age (year)		15.42 ± 1.7		15.42 ± 1.7
		Height (cm)		166.4 ± 8.5		166.5 ± 8.8
		Weight (kg)		58.6 ± 13.0		59.7 ± 13.0
		BMI (kg·m^−2^)		21.0 ± 3.6		21.4 ± 3.6

SD: standard deviation. There were no significant gender, age, and anthropometric measures differences between the normal and overdependence groups (*p* > 0.05).

**Table 2 ijerph-19-16034-t002:** Multinomial logistic regression of risk factors related to smartphone dependency. (*n* = 482).

Variable		B	S.E.	Sig.	OR (95% CI)	95% Confidence Interval for Exp (B)	
						Lower Bound	Upper Bound
Number of light PA days (≥60 min per day)	No participation in the last 7 days				1 [reference]		
	1 day	−0.03	0.35	0.92	0.97	0.49	1.90
	2 days	−0.52	0.40	0.20	0.59	0.27	1.31
	3 days	0.07	0.42	0.87	1.07	0.47	2.46
	4 days	−0.38	0.53	0.48	0.69	0.24	1.93
	5 days	−0.71	0.61	0.24	0.49	0.15	1.61
	6 days	−0.61	0.83	0.46	0.54	0.11	2.74
	Every day for the last 7 days	−1.54	0.59	0.01 *	0.21	0.07	0.69
Number of MVPA days	No participation in the last 7 days				1 [reference]		
	1 day	0.53	0.36	0.14	1.70	0.84	3.42
	2 days	0.28	0.38	0.47	1.32	0.62	2.80
	3 days	0.37	0.44	0.39	1.45	0.62	3.40
	4 days	0.80	0.59	0.17	2.23	0.70	7.06
	5 days or more	1.41	0.55	0.01 *	4.10	1.39	12.13
	No participation in the last 7 days				1 [reference]		
Number of strength training days	1 day	0.95	0.40	0.02 *	2.60	1.18	5.72
	2 days	0.18	0.44	0.69	1.19	0.50	2.84
	3 days	0.14	0.60	0.82	1.15	0.36	3.70
	4 days	0.04	0.41	0.93	1.04	0.46	2.34
	Not enough at all				1 [reference]		
Satisfaction with sleep fatigue recovery	Not enough	0.73	0.32	0.02 *	2.07	1.11	3.87
	Just enough	1.61	0.32	0.00 ***	4.99	2.66	9.34
	Enough	1.23	0.38	0.00 **	3.43	1.63	7.23
	Very enough	1.23	0.38	0.00 **	3.43	1.63	7.24

*** *p* < 0.001, ** *p* < 0.01, * *p* < 0.05, S.E: Standard Error, OR: Odd Ratio, CI: Confidence Interval, MVPA: Moderate to Vigorous Physical Activity; The reference category is the overdependence group measured by questionnaires.

## Data Availability

The datasets used and/or analyzed during the current study are available from the corresponding author upon reasonable request.

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
