# Peer review of "Association between Physical Activity, Sedentary Behavior, Satisfaction with Sleep Fatigue Recovery and Smartphone Dependency among Korean Adolescents: An Age- and Gender-Matched Study"

_ijerph, 2022, doi:10.3390/ijerph192316034_

Round 1

Reviewer 1 Report

Recommendations for Authors  

The manuscript is interesting and gives new information but needs some corrections especially in methods and results.   

Background

The previous results have well and adequately presented. It is important to study smartphone dependence of young people and its connections to everyday life all over the world. The Authors have justified they study with the problems that smartphone causes such as usage time becomes difficult to self-regulate, serious consequences for conflict with people around, physical discomfort, and difficulties in the home, school, and work-life. Also, health problems and increasing sedentary time and low-level PA are the problems. Good points! But what do we know about connections with sleep or sleep fatigue recovery? If nothing, it must be said here.

The name Tammeli is Tammelin.

Methods and results

The research design is appropriate and well described. There are two matched groups from bigger study. Also, most measures are clear. What means that BMI were classified by rounding to three decimal places?

Data Analysis should be written exactly and clearly, and variables must present clearly in descriptively in table 1. The demographic information gender, age, and anthropometrics (height, weight, and BMI), smartphone usage time (mean, SD), SB time (mean, SD), SSFR (% in different answers) could be presented in general/normal? and addiction group. Figures 1 and 2 could be after table 1 because the results are descriptive.

Multinomial regression analysis was written unclearly. There were studied association with the smartphone dependency and PA, SB, and SSFR. Not determine or predict anything.   

Result text must be changed so that it is right and clear

Table 1 is not necessary to describe the obvious results rows 159-173. Rows 174-175 are enough. After that the figure 1 and 2 results because they are descriptive.

Multinominal regression analysis should be written e,g so that it was 4 times likely to have 5 or more days MVPA in normal group than in smartphone addiction group.  There are many mistakes now.

the result from light PA days could be also present, maybe but in two groups first (1-5 days and 6-7 days) No there is one significant result: those who had light PA in 7 days are (0.20) 5 times more likely in normal group

In discussion, there is much good text but because not all results are right  

Author Response

Thank you very much for your effort.

Reviewer 2 Report

Thanks for the invitation to review this manuscript.

This is a cross-sectional study examining the association between physical activity domains, sleep fatigue recovery, and smartphone dependency among Korean adolescents with a gender and age-matched sample. Basically, the writing style (English) is ok and understandable, but could be improved, and become a bit smarter (e.g., lines 95-100).

This is an interesting study that well aligns with the scope of IJERPH. However, prior to a potential publication, several revisions should be considered. Please find below my remarks on this manuscript, that might well be addressed in a revision by the authors. I look forward to reading this revision!

MAJOR:

Introduction

·        Line 25 (and throughout the manuscript): rephrase "relatively light PA". Define your criteria for light PA in the methods' section and consistently use "light PA" - as specific as possible.

·        Likewise, be consistent with smartphone dependency and/or smartphone addiction. Both terms are used interchangeably. Please consider providing a definition for the appropriate term(s).

·        I am missing an introduction pertaining to the significance of (healthy) sleep in adolescents, as this is one of the primary outcomes. Sleep problems are mentioned in lines 59-60, but they are not as broadly and clearly introduced as PA, SB, and health problems related to smartphone addiction. Therefore, I encourage the authors to add a short paragraph beginning in line 79. Otherwise, SSFR in line 81 appears surprisingly and its relevance is not apparent to the reader.

Methods

·        Line 95: Please provide information about how adolescents and/or their legal guardians gave informed consent for participation in the survey (compare lines 321-322)!

·        Line 104: How did you assess height and weight? Were these outcomes measured (in this case, how and how valid/ reliable)? Or did you rely on self-reports? Then, why did you use three decimals? I suppose one decimal is more than sufficient!

·        Paragraph 2.3.1. Group: Information about the "Smartphone dependency scale" should be presented in another/extra paragraph, like you did for PA, SB, or SSFR. This paragraph is for the description of the groups.

·        Furthermore, important information is missing: What are the psychometric properties of this and all other scales? 

·        It remains unclear to the reader how the authors arrive at N=241 in both groups: Line 122: Please re-check for typo "n=421". Did n=241 adolescents have the maximum score (37-40 points)? Or was it n=421? Then which criteria were used to allocate them to the addiction group? Please also elaborate on your criteria for allocating n=241 out of n=9,684 adolescents to group 1. Which steps have been taken or which selection criteria have been used in this process?

·        Lines 125-130: Which scale/ which questions was/ were used to assess PA? Was there a lower limit for counting as a valid "strength training day", or "MVPA day", e.g., at least 10 minutes of exercise? 

·        Lines 131-136: Is the assessment of SB based on a single item/ question? Which unit is "average time" (line 132): minutes, hours, or hours & minutes?

·        Lines 137-141: It seems that you rather used a single item/ question instead of a questionnaire/ scale. In this case, please rephrase “questionnaire” in line 138. Again, information on the psychometric characteristics of all scales used is missing, including information on validity and/or reliability, when available. If this information is not available, this must be acknowledged in the limitations' section. Please also consider adding the questions/scales as supplementary material to the manuscript.

Data Analysis

·        Line 148: Why did you use an independent t-test instead of a 2 (groups) x 2 (for learning vs. not for learning) analysis of variance? Please consider using a 2x2 ANOVA.

·        Lines 149-150: Why did you use the Mann-Whitney U-test? And where are the corresponding results (I cannot find them)?

·        In this regard, how did you analyze "mean smartphone usage time" in figure 1 (independent t-test)?

Results

·        General: The authors compare both groups in terms of “mean smartphone usage time” and “sedentary behavior”; however, the comparisons for the remaining variables (LPA, MVPA, strength training, and SSFR are missing. Please consider adding these (as descriptive results). In my understanding, the order of results should be:

o    1. Table 1

o    2. Figure 1

o    3. Figure 2 (3&4?)

o    4. Table 2 (main results)

Again, indicate how you compared the outcomes smartphone use, SB (PA? And SSFR?), between both groups (which statistical analysis). Are your p-values corrected for multiple comparisons (e.g., Bonferroni)? Consider using MANOVA if appropriate.

·         Lines 165-166: Please indicate which statistical analysis was used for the comparison of both groups regarding anthropometrics (should be multivariate analysis of variance (MANOVA); otherwise, Bonferroni-corrections for multiple comparisons must be applied! Please indicate your approach in the "data analysis"-section, as well.

·         In line 178 it says: “smartphone addiction was unrelated to the number of relatively light PA days”. => But in table 2 there is a significant association between smartphone addiction and the number of light PA days (every day/ 7 days) with p<0.01 and OR=0.21! Please revise.

·         Table 2: It is surprising that ≥5 days of MVPA is a risk factor for smartphone addiction (OR=4.10, p=0.01), while at the same time, engaging in light PA daily is a protective factor (OR=0.21, p=0.01). I would assume that MVPA and light PA would be highly correlated, thus contradicting your finding. Therefore, please re-check the results, particularly the direction of ORs in terms of protective factors or risk factors.

·         Table 2: Be consistent regarding the order (of appearance) of your outcome variables:

1.     Light PA

2.     MVPA

3.     Strength training

4.     SSFR

Also, indicate the number of participants!

·         Lines 190-195: Please provide a measure of effect (Cohen's d, partial eta square…), as you did for the other outcomes by providing ORs.

·         Table 2 and text: Please correct "p = 0.00” => p < 0.001?

·         Lines 199-208: Again, please provide a measure of effect (Cohen's d, partial eta square…) for SB, too.

·         Figure 2: Please re-check error bars (95% CI), particularly for the "normal" group. These are too unbalanced (the lower 95% CI is much smaller compared to the upper 95% CI; and the lower 95% CI is completely missing in the third bar.

·         Line 232-234: "the number of students who participated in strength training activities for 1 day in the last 7 days was 1.7 times higher in the normal group than in the addiction group." => This is not consistent with the information provided in table 2!

Discussion

·         Lines 247-248: "Therefore, PA is intrinsically essential to reduce smartphone dependence among adolescents." This conclusion cannot be derived from the previous observations, which only describe a correlation between PA, SB, and smartphone dependence. In their statement, however, the authors make a causal inference based on the aforementioned association. Please revise.

·         Lines 225-266: From my point of view, a discussion on light PA is completely missing, but is essential given the contradictory findings with MVPA. Please add!

·         Line 292: Please rephrase "objective measures" to "device-based measures", because these are still subjective as they depend on your way of analyzing PA data, which might differ between analysts.

·         Lines 291-296: I do not think that this list of limitations is complete, yet. For instance, please refer to the questionnaire in case the scales used have not been validated before…

·         Moreover, no confounders (e.g., socioeconomic status), have been determined or used as covariates in your analyses, which would be another limitation…

·         Another limitation would be that due to the cross-sectional study design, no causal inference can be drawn.

 MINOR:

Line 19: Rephrase "this study analyzed…" => "We analyzed…"

Line 20: same

Line 22: Although you are right in avoiding passive voice, it seems legit to rephrase "The age and gender of participants […] were matched" to "Participants were matched by age and gender." interchangeably. Please consider providing a definition for the appropriate term(s).

Line 29: "in anthropometry (i.e., age, gender) matched groups" is not needed here; better use "among Korean adolescents" or "among Korean adolescents matched by age and gender".

Line 30: "Increase PA levels and days" is unclear/ not comprehensible => What is meant by PA days? Maybe better: "Increase overall PA and number of days participating in MVPA"?

Line 37: Please re-check "trillion"!

Lines 40-43: "please re-phrase this sentence; it is hard to follow in its current form"

Line 55: "In some previous studies, they revealed that the increase…" => better "Previous studies revealed that the increase…"

Lines 95-100: Please re-check the whole paragraph.

Line 95: "participated" is incorrect.

Line 96: The next sentence starts with "The purpose…", but there is no purpose! Please rephrase!

The text between lines 159-165 is redundant with the information provided in table 1 - thus can be deleted.

Lines 166-168: please rephrase "were highest" and "were lowest"; maybe like this: "Adolescents aged 17 years represented the largest group…"

Line 190: Please provide this information "Online survey" in the methods' section!

Figure 1: one decimal is sufficient! None - for readability purposes - would do as well!

Author Response

Thank you very much for your effort. 

Round 2

Reviewer 2 Report

In my view, the authors have successfully revised the manuscript according to previous comments and significantly improved its quality. Before the manuscript can be accepted, I recommend only minor improvements, mainly related to the "methods" section.

Comment 3:

Thank you for adding this information. However, my previous concern related to HOW participants gave informed consent, e.g., did they complete online informed consent before commencing the survey? Please briefly describe this procedure in the manuscript – thank you!

Comment 4:

In this case, please rephrase “the measurements of anthropometrics…”, because when participants responded to a question regarding their weight and height, you did not measure it. Maybe use: “The assessment of height and weight was based on self-reported data”.

Comment 5:

The authors elaborated on the “Smartphone dependency scale”. However, crucial information on psychometric properties is still missing (validity, reliability). At least, Cronbach’s alpha should be reported.

Comment 7:

Thank you for the explanation. Please add the exact question used for PA assessment in the survey – either in the main text or as a supplement.

Comment 8:

Please indicate in the main text that the unit of SB time is minutes. Furthermore, same as above, please add the exact question used for SB assessment in the survey – either in the main text or as a supplement.

Comment 14:

In this case, it’s fine for me. However, once you compare multiple variables between two groups, using multiple t-tests without taking measures to prevent alpha error accumulation would be inappropriate (multivariate analysis of variance (MANOVA) with Bonferroni adjustments could be applied…), though this would not alter the current results. Or have I missed the information in the main text?

Author Response

Thanks again for your insight and thorough review!
